# Online Constrained Meta-Learning: Provable Guarantees for Generalization

**Siyuan Xu & Minghui Zhu**
School of Electrical Engineering and Computer Science
The Pennsylvania State University
University Park, PA 16801
{spx5032, muz16}@psu.edu

## Abstract

Meta-learning has attracted attention due to its strong ability to learn experiences from known tasks, which can speed up and enhance the learning process for new tasks. However, most existing meta-learning approaches only can learn from tasks without any constraint. This paper proposes an online constrained meta-learning framework, which continuously learns meta-knowledge from sequential learning tasks, and the learning tasks are subject to hard constraints. Beyond existing meta-learning analyses, we provide the upper bounds of optimality gaps and constraint violations of the deployed task-specific models produced by the proposed framework. These metrics consider both the dynamic regret of online learning and the generalization ability of the task-specific models to unseen data. Moreover, we provide a practical algorithm for the framework and validate its superior effectiveness through experiments conducted on meta-imitation learning and few-shot image classification.

## 1   Introduction

In the setting of learning from multiple tasks, a shared prior is extracted from commonalities of existing tasks, resulting in improved learning efficiency and prediction accuracy for task-specific models. Conventional approaches includes multi-task learning [11], lifelong learning [46, 45, 34]. These approaches extract the prior by vanilla learning approaches on the tasks and do not explicitly involve the task-specific learning in the prior extraction, as illustrated by [28]. In contrast, meta-learning or learning-to-learn [51, 23, 47] learns a meta prior, usually a meta parameter, by evaluating and optimizing the meta-objective, which is defined by the performance of the task-specific model adapted from the prior. Meta-learning has been extended to accelerate reinforcement learning [23, 34], where the tasks are modeled by Markov decision processes.

Online meta-learning considers that the meta parameter is learned from tasks that are sequentially revealed. Papers [24, 1, 16] are shown to be no-regret with respect to the static comparator, i.e., the gap between the aggregate performance of the meta parameter sequence and that of any fixed meta parameter is sublinear in the number of revealed tasks. The online-MAML [24] and the MOML [1] adopt one-step gradient descent to adapt the task-specific parameter, and paper [16] adapts it by the ridge regression. A direct evaluation of meta-learning is the performance of the task-specific model adapted from the meta parameter on each individual task. The methods in [4, 3, 34] consider dynamic regret [56, 4], which compares the task-specific parameter adapted from the meta parameter with the optimal counterpart for each task. The dynamic regrets of all the algorithms are shown to be bounded when there are sufficiently many tasks.

In many applications of meta-learning, such as few-shot learning in computer vision [38] and robotics [43, 5, 39], a task-specific model is expected to be quickly adapted by using a small amount of

training data of a new task and generalize to the unseen data of the task. The generalization ability of the task-specific adaptation has been considered in [13, 20, 21, 31]. Most of them consider the task-specific model to be adapted using a small number of samples from the data distribution, and the generalization error is quantified by the optimality gap between the produced task-specific model and the task-specific model adapted using the best meta-parameter. In contrast, in this paper, we explore a generalization error metric, which is defined as the optimality gap between the produced task-specific model and the optimal task-specific model, based on the entire data distribution. By directly quantifying the performance of the task-specific model on the entire data (including unseen data), this metric identifies the improvement in the model's generalizability for each specific learning task when the meta-parameter is employed. Despite its significance, this generalization metric is often overlooked in most meta-learning analyses. Generalization metrics that are similar to ours are evaluated in [15, 18] and discussed in [20].

Typically, meta-learning handles learning tasks as unconstrained optimization problems. Many real-world applications such as robot learning in [14, 54, 2, 29, 42], nevertheless, are subject to safety constraints that should be rarely violated. For this case, constrained meta-learning aims to learn meta-prior for constrained learning tasks, which are formulated as constrained optimization problems. As a result, in constrained meta-learning, besides the generalization with respect to the objective risk, the generalization with respect to constraint violation is also required to be considered during the task-specific adaptation. More specifically, in the constrained learning task, each constraint is also defined by the expectation over a data distribution, and only a small amount of training data sampled from the distribution is observable during the task-specific adaptation. Therefore, the adaptation can only guarantee the constraint satisfaction over the training data, while the constraint violation over the whole data distribution is required to be quantified.

**Main contribution.** In this paper, we develop the first approach for the online constrained meta-learning problem. In particular, we consider a sequential task setting of constrained learning tasks. At each round, a learning task with hard constraints is revealed and is required to be solved, and only a small number of data for the revealed task is provided. To address the problem, we use the Follow-the-Perturbed-Leader (FTPL) approach to learn a meta-regularization at each round from the revealed tasks and apply the meta-regularization to solve the new task. The overall contributions are summarized as follows. (i) We apply constrained optimization with biased meta-regularization to adapt the task-specific parameter by using a small amount of data. Then, we optimize the meta-objective function, which is defined by the performance of the task-specific parameter on validation data. (ii) To handle the non-convex meta-objective function, we use the Follow-the-Perturbed-Leader (FTPL) to update the meta-regularization from all the available tasks for the new task. (iii) We theoretically analyze and prove the upper bounds that (a) quantify both the optimality gaps on the objective functions and the constraint violations of task-specific models; (b) quantify the dynamic regrets where the comparator is dynamic and always optimal for each task; (c) quantify the generalization errors of task-specific models to the unseen data, where only a small number of training data are given in each task. The upper bounds show that the optimality gap between the optimal meta-parameter and that learned by our method decreases with a rate of $\mathcal{O}(1/\sqrt{T})$. (iv) We develop a practical algorithm for the online constrained meta-learning that can be used effectively on large-scale problems, and empirically verify the efficacy of the proposed algorithm by conducting experiments on meta-imitation learning and few-shot robust image classification.

Table 1 compares the challenges of interests between this paper and previous works [4, 15, 24, 1, 18]. In particular, we consider the hard constraints in each learning task, the dynamic regret in the online meta-learning problem, and the generalization ability of the task-specific models to unseen data, si-

Table 1: Challenges of interests

|  | This paper | [4] | [15] | [18] | [1, 24] |
|---|---|---|---|---|---|
| Constraint | ✓ | × | × | × | × |
| Online learning | ✓ | ✓ | × | ✓ | ✓ |
| Dynamic regret | ✓ | ✓ | N/A | ✓ | × |
| Generalization | ✓ | × | ✓ | ✓ | × |

multaneously. Even if the constraints are removed, this paper provides a more general theoretical analysis than most papers, as paper [15] quantifies the empirical loss over known training data, and does not consider the generalization, and papers [24, 1] consider a static regret.

**Related works.** Optimization-based meta-learning algorithms usually learn an optimization parameter as the meta prior, and each algorithm consists of a meta-algorithm, which learns the

meta parameter, and an inner-task or within-task algorithm, which adapts to a task-specific model based on the meta parameter. Based on the type of the within-task approach, the methods can be categorized into meta-initialization approaches [41, 23, 24, 4, 35, 33] and meta-regularization approaches [17, 15, 47, 32]. In meta-initialization approaches, the within-task algorithm uses the meta parameter as the initial point and takes a few optimization steps to minimize the empirical risk of the new task. For example, the MAML [23] algorithm takes one-step gradient descent as the within-task algorithm. In meta-regularization approaches, the within-task algorithm completely solves an optimization problem, where the meta parameter serves as the bias of the regularization term for the empirical risk. In our constrained meta-learning problem, the task-specific parameter should satisfy its given constraints. In the meta-initialization approaches, as the within-task algorithm only takes a few optimization steps, even if we do optimization to reduce the constraint violation, the solution is far from feasible. On the other hand, the meta-regularization approaches fully solve the within-task. To prioritize constraint satisfaction, we combine the meta-regularization approach with hard constraints as the within-task algorithm for constrained meta-learning.

**Notations.** Denote the $l_2$ norm of vectors and the spectral norm (2-norm) of matrices by $\|\cdot\|$. Consider multiple data distributions $\mathcal{D} = \{\mathcal{D}_0, \mathcal{D}_1, \ldots, \mathcal{D}_m\}$ and multiple training datasets $\mathcal{D}^{tr} = \{\mathcal{D}_0^{tr}, \mathcal{D}_1^{tr}, \ldots, \mathcal{D}_m^{tr}\}$. We use $\mathcal{D}_i^{tr} \sim \mathcal{D}_i$ to represent that each data $z$ in the dataset $\mathcal{D}_i^{tr}$ is i.i.d sampled from the distribution $\mathcal{D}_i$, and $\mathcal{D}^{tr} \sim \mathcal{D}$ to represent $\mathcal{D}_i^{tr} \sim \mathcal{D}_i$ for all $0 \leq i \leq m$. We use $|\mathcal{D}_i^{tr}|$ to denote the number of data in the dataset $\mathcal{D}_i^{tr}$. A notation checklist is attached in Appendix A.

## 2 Problem Formulation

### 2.1 Constrained learning tasks and the sequential setting

At round $t$, an agent aims to solve a constrained learning task $\mathcal{T}_t$, which is formulated as the following constrained optimization problem:

$$\theta_t^* = \underset{\theta \in \Theta}{\operatorname{argmin}} \, \mathbb{E}_{z \sim \mathcal{D}_{0,t}} [\ell_0(\theta, z)] \text{ s.t. } \mathbb{E}_{z \sim \mathcal{D}_{i,t}} [\ell_i(\theta, z)] \leq c_{i,t}, \, i = 1, \ldots, m, \tag{1}$$

where each data point is denoted by $z = (x, y) \in \mathcal{Z}$ with $x \in \mathcal{X} \subseteq \mathbb{R}^{d_x}$ being the input and $y \in \mathcal{Y} \subseteq \mathbb{R}^{d_y}$ being its corresponding output. The loss function $\ell_0 : \mathbb{R}^d \times \mathcal{Z} \to \mathbb{R}$ is the performance metric of a model parameterized by $\theta \in \Theta \subseteq \mathbb{R}^d$ and the constraint functions $\ell_i : \mathbb{R}^d \times \mathcal{Z} \to \mathbb{R}, i = 1, \ldots, m$ evaluate the constraint metric of the model. Each performance or constraint metric $\ell_i$ is computed on the data distribution $\mathcal{D}_{i,t}$, which is a joint distribution over input-output pairs $z = (x, y)$. The optimal solution of Problem (1) is denoted as $\theta_t^*$, and the feasible set of Problem (1) is denoted as $\mathcal{K}_t \triangleq \{\theta \in \Theta \mid \mathbb{E}_{z \sim \mathcal{D}_{i,t}} [\ell_i(\theta, z)] \leq c_{i,t}, \, \forall i = 1, \ldots, m\}$. The constrained learning task $\mathcal{T}_t$ is characterized by its data distributions $\mathcal{D}_t = \{\mathcal{D}_{0,t}, \mathcal{D}_{1,t}, \ldots, \mathcal{D}_{m,t}\}$ and constraint limits $\{c_{1,t}, \ldots, c_{m,t}\}$. In general, the data distributions $\mathcal{D}_t$ are unknown to the agent. Instead, the training datasets $\mathcal{D}_t^{tr} = \{\mathcal{D}_{0,t}^{tr}, \mathcal{D}_{1,t}^{tr}, \ldots, \mathcal{D}_{m,t}^{tr}\}$ are sampled i.i.d from $\mathcal{D}_t$ and available to the agent.

Consider that a set of constrained learning tasks are revealed sequentially , i.e., task $\mathcal{T}_t$ is revealed at round $t$. In each round $t$, the agent updates a model once $\mathcal{D}_t^{tr}$ is obtained. The updated model is deployed and works on $\mathcal{T}_t$, i.e., predicting the outputs $y$ of all requested inputs $x$. In the next round, the procedure repeats.

Since the data distributions $\mathcal{D}_t$ are unknown, the agent cannot completely solve Problem (1) and obtain $\theta_t^*$. A common solution is to replace the data distribution $\mathcal{D}_{i,t}$ in Problem (1) by the sampled dataset $\mathcal{D}_{i,t}^{tr}$ and solving the problem for each task $\mathcal{T}_t$. However, when the sample number in $\mathcal{D}_{i,t}^{tr}$ is small, separately solving $\mathcal{T}_t$ with $\mathcal{D}_{i,t}^{tr}$ leads to limited performance. Similar to [4, 15], in this paper, the learning algorithm is expected to improve the performance of the solution for task $\mathcal{T}_t$ by exploiting the correlation between the task $\mathcal{T}_t$ and all previous tasks $\{\mathcal{T}_1, \cdots, \mathcal{T}_{t-1}\}$.

### 2.2 Metrics of a learning algorithm

Consider a learning algorithm produces a parameter sequence $\{\theta_1, \cdots, \theta_T\}$ for the task sequence $\{\mathcal{T}_1, \cdots, \mathcal{T}_T\}$. At round $t$, the optimality gap $R_{0,t}$ of $\theta_t$ is defined as $R_{0,t}(\theta_t) = \max\{\mathbb{E}_{z \sim \mathcal{D}_{0,t}} [\ell_0(\theta_t, z) - \ell_0(\theta_t^*, z)], 0\}$. The violation for the $i$-th constraint denoted by $R_{i,t}$ is defined as: $R_{i,t}(\theta_t) = \max\{\mathbb{E}_{z \sim \mathcal{D}_{i,t}} [\ell_i(\theta_t, z)] - c_{i,t}, 0\}, i = 1, \ldots, m$. The task-averaged optimality gap (TAOG) denoted by $\bar{R}_{0,[1:T]}$ and the task-averaged constraint violation (TACV) denoted

by $\bar{R}_{i,[1:T]}$ of the algorithm after $T$ rounds are defined as

$$\bar{R}_{0,[1:T]} = \frac{1}{T}\sum_{t=1}^{T} R_{0,t}(\theta_t), \quad \bar{R}_{i,[1:T]} = \frac{1}{T}\sum_{t=1}^{T} R_{i,t}(\theta_t), \; i = 1, \ldots, m. \tag{2}$$

Under the sequential task setting, the goal of a learning algorithm is to minimize the TAOG and the TACV of the parameter sequence $\{\theta_1, \cdots, \theta_T\}$. The definitions of the TAOG and the TACV consider both the generalization ability of $\theta_t$ to the unseen data of each revealed task $\mathcal{T}_t$, and a dynamic notion of the optimality regret. First, the optimality gap $R_{0,t}$ and the constraint violation $R_{i,t}$ are defined by the expectations over the inaccessible data distribution $D_{i,t}$, instead of the empirical risk over the given dataset $D_{i,t}^{tr}$ that is used in online meta-learning analysis [4, 35]. Thus, the TAOG and the TACV can be used to quantify the generalization errors. It is more challenging to minimize them, as $\theta_t$ is expected to generalize to the unseen data in $D_{i,t}$ while only the training datasets $D_{i,t}^{tr}$ are given, and their size could be small. Second, unlike the fixed comparator used in the FTML algorithm [24], the comparator $\theta_t^*$ is dynamic and always optimal for each task $\mathcal{T}_t$. As pointed out in [4, 34, 56], one cannot achieve the dynamic regret sublinear in $T$ for either online learning or online meta-learning.

### 2.3 Task dissimilarity

As mentioned in Section 2.1, we expect that the performance produced by the learning algorithm, i.e., the TAOG and the TACV, are improved as the similarity and correlation among the task sequence $\{\mathcal{T}_1, \cdots, \mathcal{T}_T\}$ are higher. Similar to [4, 15, 34], we define the dissimilarity between tasks by the distance between the optimal parameters of tasks. Given the optimal task-specific parameters $\{\theta_1^*, \cdots, \theta_T^*\}$ for the task sequence $\{\mathcal{T}_1, \cdots, \mathcal{T}_T\}$, the average distance between the parameter $\phi$ and the optimal task-specific parameters is defined by $\mathcal{D}ist(\phi, \mathcal{T}_{1:T}) \triangleq \sqrt{\frac{1}{T}\sum_{t=1}^{T}\frac{1}{2}\|\theta_t^* - \phi\|^2}$. The dissimilarity of $\{\mathcal{T}_1, \cdots, \mathcal{T}_T\}$ is defined as $\mathcal{S}^*(\mathcal{T}_{1:T}) \triangleq \min_\phi \mathcal{D}ist(\phi, \mathcal{T}_{1:T})$. When the disclosure of $\{\mathcal{T}_1, \cdots, \mathcal{T}_T\}$ is stationary, i.e., the task $\mathcal{T}_t$ at each $t$ is sampled from the same probability distribution $p(\mathcal{T})$, the task dissimilarity of the task distribution $p(\mathcal{T})$ is defined as $\mathcal{S}^*(p(\mathcal{T})) \triangleq \sqrt{\min_\phi \mathbb{E}_{\mathcal{T}_t \sim p(\mathcal{T})}\left[\frac{1}{2}\|\theta_t^* - \phi\|^2\right]}$.

## 3 Online Constrained Meta-Learning Algorithm

In this section, to handle the sequential constrained learning tasks formulated in Section 2.1, we propose the online constrained meta-learning framework, which contains the meta-parameter update in Section 3.2 and the task-specific adaptation from the meta-parameter in Section 3.3.

### 3.1 Online constrained meta-learning setting.

Online constrained meta-learning aims to handle the problem given in Section 2.1, where the sample numbers of the datasets for each task, i.e., $|\mathcal{D}_{i,t}^{tr}|$, are limited and the agent is required to quickly adapt to a new task once the task is revealed. At round $t$, the task-specific parameter $\theta_t$ is updated by a within-task algorithm, and the meta parameter $\phi_t$ is updated by a meta-algorithm. In particular, in the first step, the agent adapts the meta parameter $\phi_t$ to the task-specific parameter $\theta_t$ for task $\mathcal{T}_t$, by using a within-task algorithm $\mathcal{A}lg$ with the training datasets $\mathcal{D}_t^{tr} = \{\mathcal{D}_{0,t}^{tr}, \cdots, \mathcal{D}_{m,t}^{tr}\}$ and the constraint limits $\{c_{1,t}, \ldots, c_{m,t}\}$. Next, the agent uses all available tasks $\mathcal{T}_1, \ldots, \mathcal{T}_t$ to update the meta parameter to $\phi_{t+1}$ by a meta-algorithm, which will be used in the task-specific adaptation for the next task $\mathcal{T}_{t+1}$ at round $t+1$. Our task-specific adaptation algorithm $\mathcal{A}lg$ and meta-algorithm are discussed in the following.

### 3.2 Task-specific adaptation.

At round $t$, the task-specific parameter $\theta_t$ is adapted from the meta parameter $\phi_t$ to task $\mathcal{T}_t$ by the within-task algorithm $\mathcal{A}lg$. In the meta-initialization approaches [41, 23, 24], the within-task algorithm $\mathcal{A}lg$ only takes a few optimization steps toward reducing the empirical risk and the constraint violation, and thus the solution is far from being feasible for Problem (1). To prioritize constraint satisfaction, we employ the meta-regularization approach with hard constraints. In particular, the

task-specific adaptation from the meta-parameter $\phi_t$ is defined by constrained optimization with the $\phi_t$-biased regularization, i.e., $\theta_t = \mathcal{A}lg(\lambda, \phi_t, \mathcal{D}_t^{tr})$ is defined by

$$\underset{\theta \in \Theta}{\operatorname{argmin}} \frac{1}{|\mathcal{D}_{0,t}^{tr}|} \sum_{z \in \mathcal{D}_{0,t}^{tr}} \ell_0(\theta, z) + \frac{\lambda}{2} \|\theta - \phi_t\|^2 \text{ s.t. } \frac{1}{|\mathcal{D}_{i,t}^{tr}|} \sum_{z \in \mathcal{D}_{i,t}^{tr}} \ell_i(\theta, z) \le c_{i,t}, \ i = 1, \dots, m, \quad (3)$$

where the regularization weight $\lambda > 0$. By the algorithm in (3), the task-specific parameter $\theta_t$ satisfies the constraints over the training datasets $\mathcal{D}_{i,t}^{tr}$. The objective function includes the empirical loss defined on the training dataset $\mathcal{D}_{0,t}^{tr}$ and is regularized by a biased term $\frac{\lambda}{2} \|\theta - \phi_t\|^2$ [47, 15], which penalizes the deviation from the meta parameter $\phi_t$. The feasible set of the optimization problem in (3) is denoted as $\mathcal{K}_t^{tr} \triangleq \left\{ \theta \in \Theta \mid \frac{1}{|\mathcal{D}_{i,t}^{tr}|} \sum_{z \in \mathcal{D}_{i,t}^{tr}} \ell_i(\theta, z) \le c_{i,t}, \ \forall i = 1, \dots, m \right\}$. In the following sections, we consider that, the numbers of the data sampling from $\mathcal{D}_{0,t}$, i.e., $|\mathcal{D}_{0,t}^{tr}|$, are the same for all $t$, and is denoted as $|\mathcal{D}_0^{tr}|$; the numbers of the data sampling from $\mathcal{D}_{i,t}$, i.e., $|\mathcal{D}_{i,t}^{tr}|$ are the same for each $i$ and $t$, and is denoted as $|\mathcal{D}_+^{tr}|$.

### 3.3 Meta-parameter update.

After the model parameterized by $\theta_t$ is deployed, the agent obtains the dataset $\mathcal{D}_t^{val} = \{\mathcal{D}_{0,t}^{val}, \cdots, \mathcal{D}_{m,t}^{val}\}$ by sampling from the data distribution $\mathcal{D}_t = \{\mathcal{D}_{0,t}, \cdots, \mathcal{D}_{m,t}\}$. The performance evaluation of the model with parameter $\theta$ is defined as $\mathcal{L}^{val}(\theta, \mathcal{D}_{0,t}^{val}) = \frac{1}{|\mathcal{D}_{0,t}^{val}|} \sum_{z \in \mathcal{D}_{0,t}^{val}} \ell_0(\theta, z)$. Here, we consider that $|\mathcal{D}_{0,t}^{val}|$ is the same for all $t$ and is denoted as $|\mathcal{D}_0^{val}|$.

In online meta-learning, at each round $t$, the meta-parameter is updated by using an online learning algorithm to the meta-objective functions on all revealed tasks $\{\mathcal{T}_1, \cdots, \mathcal{T}_t\}$. In our meta-learning problem, as the agent has collected the validation data $\{\mathcal{D}_1^{val}, \cdots, \mathcal{D}_t^{val}\}$, we can evaluate the performance of the task-specific parameter $\theta_t$ for each task, then the meta-objective function at round $t$ is defined as $\sum_{t'=1}^{t} \mathcal{L}^{val}(\mathcal{A}lg(\lambda, \phi, \mathcal{D}_{t'}^{tr}), \mathcal{D}_{0,t'}^{val})$, where $\mathcal{A}lg$ is defined in (3).

Papers [4, 24] solve online meta-learning and update the meta-parameter by using the Follow-the-Leader (FTL) [26] to their meta-objective functions. However, the FTL requires the objective function to be strongly convex, and cannot be used in our problem, where the meta-objective function is non-convex. Therefore, in this paper, we use the Follow-the-Perturbed-Leader (FTPL) [50] to solve the problem, which is the first time that the FTPL is applied to online meta-learning. At round $t$, the FTPL optimizes the meta-objective function with perturbed terms over all revealed tasks to obtain the meta parameter $\phi_{t+1}$ for task $\mathcal{T}_{t+1}$. In particular, at round $t$, the meta parameter $\phi_{t+1}$ is obtained by solving the following optimization problem:

$$\phi_{t+1} = \underset{\phi \in \Theta}{\operatorname{argmin}} \sum_{t'=1}^{t} \mathcal{L}^{val}(\mathcal{A}lg(\lambda, \phi, \mathcal{D}_{t'}^{tr}), \mathcal{D}_{0,t'}^{val}) - \sigma_t^\top \phi, \quad (4)$$

where $\sigma_t \in \mathbb{R}^d$ is the random perturbed vector and its components $\{\sigma_{t,j}\}_{j=1}^{d}$ is i.i.d sampled from the exponential distribution with a parameter $\eta > 0$ at each $t$. As $\mathcal{A}lg(\lambda, \phi, \mathcal{D}_{t'}^{tr})$ included in (4) is the optimal solution of the constrained optimization in (3), Problem (4) is a constrained bilevel optimization problem [55].

---

**Algorithm 1** Online Constrained Meta-Learning Framework

---

**Require:** Regularization weight $\lambda > 0$; Initial meta parameter $\phi_1$; Perturbed parameter $\eta$.
 1: **for** $t = 1, \cdots, T$ **do**
 2:    Sample the training datasets $\mathcal{D}_t^{tr}$ from the distributions $\mathcal{D}_t$ for task $\mathcal{T}_t$
 3:    Update and deploy the task-specific parameter $\theta_t = \mathcal{A}lg(\lambda, \phi_t, \mathcal{D}_t^{tr})$ defined in (3) for task $\mathcal{T}_t$
 4:    Sample the evaluation dataset $\mathcal{D}_{0,t}^{val}$ from the distributions $\mathcal{D}_{0,t}$
 5:    Generate the random perturbed vector $\sigma_t \in \mathbb{R}^d$ by i.i.d sampling: $\{\sigma_{t,j}\}_{j=1}^{d} \sim \operatorname{Exp}(\eta)$
 6:    Update the meta parameter $\phi_{t+1}$ by solving (4)
 7: **end for**
 8: Return $\{\theta_1, \cdots, \theta_T\}$ for tasks $\{\mathcal{T}_1, \cdots, \mathcal{T}_T\}$

---

### 3.4 Algorithm statement.

The online constrained meta-learning algorithm is formally stated in Algorithm 1. Note that line 3 is a deterministic convex optimization problem, which has been widely studied and solved by [10, 6]. The optimization problem (4) in line 6 is a deterministic constrained bilevel optimization problem, which can be solved by the algorithm in [55]. The implementation details are included in our proposed practical algorithm (Algorithm 2 in Appendix B).

## 4 Theoretical Results

In this section, we derive the upper bounds of the TAOG and the TACV produced by Algorithm 1, which characterize the regrets of the generalization errors of the models deployed for the sequential tasks. First, we study the generalization error bound produced by our within-task algorithm (3) for a single constrained learning task (Section 4.1). Next, we show the upper bounds of the TAOG and the TACV when the meta-parameter can be arbitrarily selected (Section 4.1). Finally, we obtain the theoretical bounds of the TAOG and the TACV, when the meta-parameter is not arbitrarily selected bu produced by Algorithm 1 (Section 4.2).

Before the result statements, we introduce some definitions and the required assumptions about the constraint qualifications for the optimization problems in (1) and (3). The constraint qualification assumptions are usually used in constraint optimization analyses [7, 19].

We consider an optimization problem with inequality constraints: $\min_x g(x)$ s.t. $h_i(x) \leq 0$, $i \in I \triangleq \{1, \ldots, m\}$ and denote it as (P). Denote the feasible set of (P) as $\mathcal{K} \triangleq \{x \mid h_i(x) \leq 0, \ \forall i \in I\}$ and $I(x) \triangleq \{i \in I \mid h_i(x) = 0\}$ for $x \in \mathcal{K}$.

**Definition 1.** *The Linear Independence Constraint Qualifications (LICQ) holds for (P), if vectors $\{\nabla h_i(x) \mid i \in I(x)\}$ are linearly independent for any $x \in \mathcal{K}$.*

**Definition 2.** *Slater's condition (SC) holds for (P) with the margin $\mathcal{C} > 0$, if there exists $\bar{x}$ such that $h_i(\bar{x}) - \mathcal{C} \leq 0$, $\forall i \in I$.*

**Assumption 1** (Constraint qualifications). *(i) There exists a compact set with diameter $\mathcal{B} > 0$ such that the following properties hold for each $t$: for the given data distribution $\mathcal{D}_t$, the feasible set $\mathcal{K}_t$ of the optimization problem in (1) is included in the compact set; with probability $1$, for $\mathcal{D}_t^{tr}$ sampled from $\mathcal{D}_t$, the feasible set $\mathcal{K}_t^{tr}$ of the optimization problem in (3) is included in the compact set.*

*(ii) There exists $\mathcal{C} > 0$ such that the following properties hold: for the given data distribution $\mathcal{D}_t$, the SC holds for the optimization problem in (1) with the margin $\mathcal{C}$; with probability $1$, for $\mathcal{D}_t^{tr}$ sampled from $\mathcal{D}_t$, the SC holds for the optimization problem in (3) with the margin $\mathcal{C}$.*

*(iii) For the given data distribution $\mathcal{D}_t$, the LICQ holds for the optimization problem in (1).*

**Remark 1.** *A sufficient condition for part (i) of Assumption 1 is that $\Theta$ is a compact set.*

We also require the following assumptions on the loss function and the constraint functions, which are standard in the analysis of meta-learning problems [4, 15, 24, 20].

**Assumption 2** (Function properties). *(i) For any $z \in \mathcal{Z}$, the loss function $\ell_0(\cdot, z)$ and the constraint functions $\ell_1(\cdot, z), \cdots, \ell_m(\cdot, z)$ are twice continuously differentiable. (ii) For any $z \in \mathcal{Z}$, $\ell_0(\cdot, z)$ is $L_0$-Lipschitz, i.e., $\|\ell_0(w, z) - \ell_0(u, z)\| \leq L_0 \|w - u\|$ for any $w, u \in \Theta$. (iii) For any $z \in \mathcal{Z}$, $\ell_0(\cdot, z)$ is $\rho$-smooth, i.e., $\|\nabla \ell_0(w, z) - \nabla \ell_0(u, z)\| \leq \rho \|w - u\|$ for any $w, u \in \Theta$. (iv) For any $z \in \mathcal{Z}$ and $1 \leq i \leq m$, $\ell_i(\cdot, z)$ is $L_c$-Lipschitz, i.e., $\|\ell_i(w, z) - \ell_i(u, z)\| \leq L_c \|w - u\|$ for any $w, u \in \Theta$. (v) For any $z \in \mathcal{Z}$ and $0 \leq i \leq m$, $\ell_i(\cdot, z)$ is convex. (vi) For any $t \in \{1, \cdots, T\}$, $z \in \mathcal{Z}$, $w \in \Theta$, and $1 \leq i \leq m$, $\ell_i(w, z) - c_{i,t}$ are bounded by $M$.*

### 4.1 Generalization of constrained learning with biased regularization

We begin with the generalization error bound on a single constrained learning task $\mathcal{T}_t$, which is produced by the within-task algorithm (3) with an arbitrarily given meta-parameter $\phi_t$. As stated in (3), the algorithm replaces all the expectations in (1) by the sample averages oven the given datasets and adds a regularization term with the bias $\phi_t$ to the objective function.

**Proposition 1.** *Suppose that Assumptions 1 and 2 are satisfied. For any given meta-parameter $\phi_t$ and regularization weight $\lambda > 0$, the following bounds hold for task*

$$\mathcal{T}_t: \quad \mathbb{E}_{\mathcal{D}_t^{tr} \sim \mathcal{D}_t} \left[ R_{0,t}(\mathcal{A}lg(\lambda, \phi_t, \mathcal{D}_t^{tr})) \right] \quad \leq \quad \frac{\lambda}{2} \| \theta_t^* - \phi_t \|^2 \;+\; \mathcal{O}\left( \frac{\ln |\mathcal{D}_0^{tr}|}{\lambda |\mathcal{D}_0^{tr}|} + \sqrt{\frac{\ln |\mathcal{D}_+^{tr}|}{|\mathcal{D}_+^{tr}|}} \right), \quad and$$

$$\mathbb{E}_{\mathcal{D}_t^{tr} \sim \mathcal{D}_t} \left[ R_{i,t}(\mathcal{A}lg(\lambda, \phi_t, \mathcal{D}_t^{tr})) \right] \leq \mathcal{O}\left( \sqrt{\frac{\ln |\mathcal{D}_+^{tr}|}{|\mathcal{D}_+^{tr}|}} \right), \forall i = 1, \ldots, m.$$

Proposition 1 shows the generalization error bounds of the loss function $\ell_0$ and the constraint violation for each $\ell_i$. The coefficients of the notations $\mathcal{O}$ and the proofs are shown in Propositions 3 and 4 of Appendix E. As the objective function in (3) includes the biased regularization term $\frac{\lambda}{2} \| \theta - \phi_t \|^2$, where the bias serves as the prior estimation for the solution, the generalization error bound includes a term of $\frac{\lambda}{2} \| \theta_t^* - \phi_t \|^2$, which is decreasing as the estimation $\phi_t$ is more accurate. The objective function and the constraint functions include the sample averages over the training dataset $\mathcal{D}_t^{tr}$. As the numbers of data points ($|\mathcal{D}_0^{tr}|$ and $|\mathcal{D}_+^{tr}|$) are larger, the sample average approximation is more accurate, and then the generalization error is smaller.

The generalization error bounds shown in Proposition 1 motivate learning a good meta-parameter $\phi_t$ by meta-learning. Next, we study the generalization error bounds for the task sequence $\{\mathcal{T}_1, \cdots, \mathcal{T}_T\}$, when the meta-parameter $\phi$ is arbitrary and fixed. Proposition 2 shows the upper bounds of the TAOG and the TACV for any given meta-parameter $\phi$, i.e., the task-specific parameter $\theta_t$ in (2) is computed by $\theta_t = \mathcal{A}lg(\lambda, \phi, \mathcal{D}_t^{tr})$ for all $t$.

**Proposition 2.** *Suppose that Assumptions 1 and 2 are satisfied. Consider the task sequence $\{\mathcal{T}_1, \cdots, \mathcal{T}_T\}$, for any given meta-parameter $\phi$, choose the regularization parameter $\lambda = \frac{2\sqrt{d}(\rho \mathcal{B} + L_0 \sqrt{\ln |\mathcal{D}_0^{tr}|})}{\mathcal{D}ist(\phi, \mathcal{T}_{1:T}) \sqrt{|\mathcal{D}_0^{tr}|}}$. Then, the following bounds hold:*

$$\frac{1}{T} \sum_{t=1}^T \mathbb{E}_{\mathcal{D}_t^{tr} \sim \mathcal{D}_t} \left[ R_{0,t}(\mathcal{A}lg(\lambda, \phi, \mathcal{D}_t^{tr})) \right] \leq \mathcal{O}\left( \mathcal{D}ist(\phi, \mathcal{T}_{1:T}) \sqrt{\frac{\ln |\mathcal{D}_0^{tr}|}{|\mathcal{D}_0^{tr}|}} + \sqrt{\frac{\ln |\mathcal{D}_+^{tr}|}{|\mathcal{D}_+^{tr}|}} \right), \qquad and$$

$$\frac{1}{T} \sum_{t=1}^T \mathbb{E}_{\mathcal{D}_t^{tr} \sim \mathcal{D}_t} \left[ R_{i,t}(\mathcal{A}lg(\lambda, \phi, \mathcal{D}_t^{tr})) \right] \leq \mathcal{O}\left( \sqrt{\frac{\ln |\mathcal{D}_+^{tr}|}{|\mathcal{D}_+^{tr}|}} \right), \forall i = 1, \ldots, m.$$

The coefficients of the notations $\mathcal{O}$ and the proofs are shown in Proposition 5 of Appendix E. Following the result in Proposition 2, as the average distance between the meta-parameter $\phi$ and the optimal task-specific parameters $\{\theta_1^*, \cdots, \theta_T^*\}$, i.e., $\mathcal{D}ist(\phi, \mathcal{T}_{1:T})$, becomes small, the upper bound of the TAOG is small. By minimizing the distance $\mathcal{D}ist(\phi, \mathcal{T}_{1:T})$ over $\phi$, we can get the optimal upper bound of the TAOG:

$$\frac{1}{T} \sum_{t=1}^T \mathbb{E}_{\mathcal{D}_t^{tr} \sim \mathcal{D}_t} \left[ R_{0,t}(\mathcal{A}lg(\lambda, \phi^*, \mathcal{D}_t^{tr})) \right] \leq \mathcal{O}\left( \mathcal{S}^*(\mathcal{T}_{1:T}) \sqrt{\frac{\ln |\mathcal{D}_0^{tr}|}{|\mathcal{D}_0^{tr}|}} + \sqrt{\frac{\ln |\mathcal{D}_+^{tr}|}{|\mathcal{D}_+^{tr}|}} \right), \quad (5)$$

when the meta-parameter is selected by $\phi^* = \arg\min_\phi \mathcal{D}ist(\phi, \mathcal{T}_{1:T})$. Note that the optimal upper bound is not achievable since the optimal task-specific solution $\theta_t^*$ is not achievable for each $t$. In the next section, we show that the sequence of meta-parameters $\phi_t$ produced by Algorithm 1 produce a comparable performance as the optimal meta-parameter $\phi^*$.

### 4.2 Generalization of the online constrained meta-learning algorithm

Consider the meta-parameter is updated by (4) in Algorithm 1. As we apply the FTPL algorithm to the meta-objective function for the online constrained meta-learning problem, the upper bounds of the TAOG and the TACV produced by Algorithm 1 are shown in Theorem 1 and Corollary 1. As the algorithm to compute $\theta_t$ is stochastic, which depends on the task sequence and the sampling of the training data $\mathcal{D}_t^{tr}$ and evaluation data $\mathcal{D}_{0,t}^{val}$, we denote the expectation of the optimality gap and the constraint violations $R_{i,t}(\theta_t)$ as $\mathbb{E}[R_{i,t}(\theta_t)]$, and the expectations of the TAOG and the TACV are denoted as $\mathbb{E}[\bar{R}_{i,[1:T]}] = \mathbb{E}\left[ \frac{1}{T} \sum_{t=1}^T R_{i,t}(\theta_t) \right]$ for all $i = 0, \ldots, m$.

**Theorem 1.** *Suppose that Assumptions 1 and 2 are satisfied. Suppose that $\Theta$ is included in a compact cube with edge of length $D_l$, i.e. $\| \phi \|_\infty \leq D_l$ for any $\phi \in \Theta$. Choose the regularization parameter $\lambda = \frac{2\sqrt{d}(\rho \mathcal{B} + L_0 \sqrt{\ln |\mathcal{D}_0^{tr}|})}{\mathcal{S}^*(\mathcal{T}_{1:T}) \sqrt{|\mathcal{D}_0^{tr}|}}$ and the perturbed parameter $\eta = \frac{1}{L_0 \sqrt{dT}}$. For the task*

*sequence* $\{\mathcal{T}_1, \cdots, \mathcal{T}_T\}$, *the following bounds hold for the TAOG and the TACV of Algorithm 1:* $\mathbb{E}[\bar{R}_{0,[1:T]}] \leq \mathcal{O}(\mathcal{S}^*(\mathcal{T}_{1:T})\sqrt{\frac{\ln|\mathcal{D}_0^{tr}|}{|\mathcal{D}_0^{tr}|}} + \sqrt{\frac{\ln|\mathcal{D}_+^{tr}|}{|\mathcal{D}_+^{tr}|}} + \sqrt{\frac{\ln|\mathcal{D}_0^{val}|}{|\mathcal{D}_0^{val}|}} + \frac{1}{\sqrt{T}})$, *and* $\mathbb{E}[\bar{R}_{i,[1:T]}] \leq \mathcal{O}\left(\sqrt{\frac{\ln|\mathcal{D}_+^{tr}|}{|\mathcal{D}_+^{tr}|}}\right)$, $\forall i = 1, \ldots, m$.

**Corollary 1.** *If the task $\mathcal{T}_t$ at each $t$ is sampled from a fixed distribution $p(\mathcal{T})$, we choose $\lambda = \frac{2\sqrt{d}(\rho\mathcal{B} + L_0\sqrt{\ln|\mathcal{D}_0^{tr}|})}{\mathcal{S}^*(p(\mathcal{T}))\sqrt{|\mathcal{D}_0^{tr}|}}$, then $\mathcal{S}^*(\mathcal{T}_{1:T})$ in the upper bound of $\mathbb{E}[\bar{R}_{0,[1:T]}]$ shown in Theorem 1 can be replaced by the fixed constant $\mathcal{S}^*(p(\mathcal{T}))$.*

The coefficients of the notations $\mathcal{O}$ and the proofs are shown in Appendices E and F. In the online constrained meta-learning problem, as stated in Algorithm 1, after the task-specific parameter $\theta_t$ for task $\mathcal{T}_t$ is deployed, the agent is usually able to sample the dataset $\mathcal{D}_0^{val}$ and $|\mathcal{D}_0^{val}| \gg |\mathcal{D}_0^{tr}|$. In this case, we can ignore the term $\sqrt{\frac{\ln|\mathcal{D}_0^{val}|}{|\mathcal{D}_0^{val}|}}$ in the upper bound of $\mathbb{E}[\bar{R}_{0,[1:T]}]$ in Theorem 1. We compare the upper bound of the TAOG by Algorithm 1 with the unachievable optimal upper bound shown in (5), where the term $\sqrt{\frac{\ln|\mathcal{D}_0^{val}|}{|\mathcal{D}_0^{val}|}}$ is ignored. The only additional term included in Theorem 1 is $\mathcal{O}(\frac{1}{\sqrt{T}})$. It implies that the regret of the optimality gap between the meta-parameter $\phi_t$ updated by Algorithm 1 and the optimal meta-parameter $\phi^*$ used in (5) is sublinear with respect to the task number $T$.

In particular, when the constraint is ignored from Problem (1), the upper bound of the TAOG reduces to $\mathbb{E}[\bar{R}_{0,[1:T]}] \leq \mathcal{O}(\mathcal{S}^*(p(\mathcal{T}))\sqrt{\frac{\ln|\mathcal{D}_0^{tr}|}{|\mathcal{D}_0^{tr}|}} + \frac{1}{\sqrt{T}})$, which has the same order as the upper bound shown in [18] with respect to the task dissimilarity $\mathcal{S}^*(p(\mathcal{T}))$, the data number $|\mathcal{D}_0^{tr}|$, and task number $T$.

Corollary 1 considers that the revealed tasks are sampled from a static task distribution. In this case, when $T$ is sufficiently large, Algorithm 1 degenerates into an offline constrained meta-learning algorithm. If we further ignore the constraints in Problem (1), the bound in Corollary 1 of the TAOG holds the same order as the upper bound shown in [15].

## 5 Applications

### 5.1 Meta imitation learning

Meta-imitation Learning [25, 40] has been developed as a powerful tool for a learner to finish a new task by observing the demonstrations of an expert. However, existing methods do not consider any constraint, e.g., collision avoidance, for the new task. In this experiment, we apply the constrained meta-learning algorithm to a constrained meta-imitation learning problem where the learner learns to write letters in a cluttered environment [30, 29]. At each round $t$, the expert writes a different letter in a free space. The learner can observe the locations and velocities of a few points of the letter and is asked to write the same letter in a cluttered environment where the obstacle is known. The learner learns the policy by using constrained learning, where the loss function is defined by the tracking error of the learned policy, and the constraint is defined by a collision metric. Note that the expert performs in the free space, and the learner performs in the cluttered environment. The additional obstacle makes the problem more difficult. To the best of our knowledge, it is the first time to study

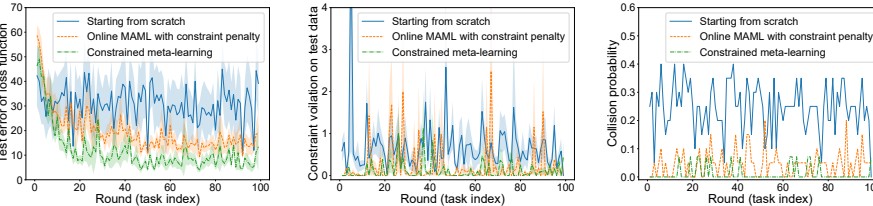

Figure 1: Results of few-shot imitation learning. **Left**: Loss on test data; **Middle**: Constrained violation metric on test data. **Right**: Collision probability.

the problem. We provide the details of the problem formulation and the implementation setting in Appendix C.1.

We compare the proposed method with two benchmarks: (a) starting from scratch; (b) online MAML [24] with constraint penalty. In (a), the optimization problem in (3) is solved from a random initial parameter. In (b), we add a weighted penalty term for constraint violation to the loss function of the online MAML. For each task, the three methods share the data points.

**Few-shot imitation learning.** Consider that the learner collects the locations and velocities on a few points (few-shot data) from the expert for each task. Fig. 1 compares the proposed constrained meta-learning with the selected benchmarks. Fig. 1 Left shows that our approach has the fastest learning rate, i.e., its test error is smaller than 20 at around $T = 10$, while the online MAML achieves that at around $T = 30$, and the test error of starting from scratch is always above 20. Moreover, our approach has the best learning accuracy, i.e., the steady-state test error of our approach is much smaller than those of the other two methods. In terms of constraint violation, our approach achieves the lowest collision probability in Fig. 1 Right and the lowest constraint violation metric in Fig. 1 Middle.

**Improve full-shot imitation learning by meta-learning.** We consider that the learner can collect the locations and velocities on sufficient points (full-shot data) from the expert for each task, and test the proposed method and the selected benchmarks. In this experiment, we show that meta-learning can improve the performance of constrained learning, even if suf-

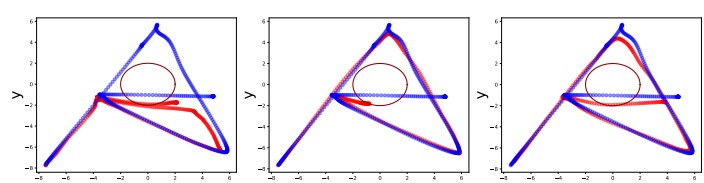

Figure 2: Learned trajectories for full-shot imitation learning. **Left**: Starting from scratch; **Middle**: MAML with constraint penalty; **Right**: Constrained meta-learning. The blue line is the demonstration trajectory, the red line is the learned trajectory, the red circle represents the obstacle.

ficient data can be accessed by the learner. As shown in Fig. 2, our method can achieve superior performance than that in the two benchmark approaches (only our method can completely draw the shape of the letter). Fig. 3 shows that our method achieves a comparable adaptation time to online-MAML while outperforming online-MAML in terms of test error and collision avoidance. Both our method and online-MAML much outperform starting from scratch in terms of test error, collision avoidance, and adaptation time.

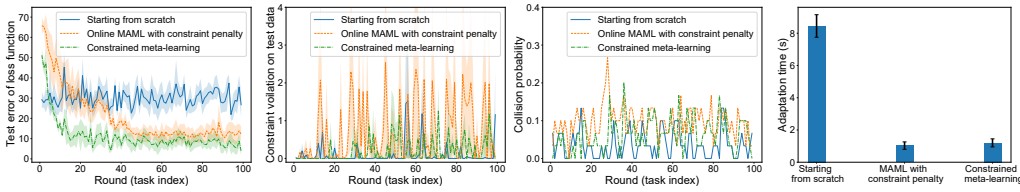

Figure 3: Results of full-shot imitation learning. **Left**: Loss on test data; **Middle left**: Constrained violation metric on test data; **Middle right**: Collision probability; **Right**: Adaptation time.

## 5.2 Few-shot image classification with robustness

Meta-learning has been widely used for few-shot learning. Few-shot learning aims to train a model with only a few data samples, which is widely solved from the perspective of meta-learning by learning prior knowledge from multiple tasks. Consider a problem of few-shot image classification with robust learning [57]. In particular, after it is updated by a few data points, we require that the task-specific model has high accuracy on clean test data and is also robust to adversarial attacks on the test data. Robust learning for a single task can be formulated by a constrained optimization problem [12]. Correspondingly, in this experiment, the robust few-shot image classification is formulated as a constrained meta-learning problem and is solved by the proposed algorithm. The details of the problem formulation and the implementation setting are shown in Appendix C.2.

To evaluate the performance of the proposed algorithm, we test the task-specific model on two sets of each dataset: (a) the clean test dataset; (b) the corrupted test dataset which is obtained by adding perturbations on the clean test dataset through the Projected Gradient Descent (PGD) method [37]. Correspondingly, we show the test accuracy on (a) as clean accuracy; the test accuracy on (b) as PGD accuracy. Moreover, we use Balance Score (B-score) [57] as a metric to evaluate the comprehensive performance of the model, which is defined as B-score = 2 × (CA × PA)/(CA + PA) with CA and PA denoting clean accuracy and PGD

Table 2: Clean accuracy (abbreviated as "Clean Acc.") and PGD accuracy (abbreviated as "PGD Acc.") on the mini-ImageNet dataset for 5-way 5-shot and 5-way 1-shot learning.

|  | Method | Clean Acc. | PGD Acc. | B-score |
|---|---|---|---|---|
| 1-shot | MAML + CP | $40.78 \pm 0.75$ | $23.91 \pm 0.67$ | $29.83 \pm 0.43$ |
|  | MAML + MOML | $39.23 \pm 0.76$ | $25.80 \pm 0.67$ | $31.12 \pm 0.70$ |
|  | ProtoNet + CP | $38.65 \pm 0.72$ | $23.10 \pm 0.65$ | $28.67 \pm 0.67$ |
|  | ProtoNet + MOML | $35.06 \pm 0.70$ | $27.24 \pm 0.65$ | $30.51 \pm 0.66$ |
|  | BOIL + CP | $40.44 \pm 0.79$ | $25.94 \pm 0.69$ | $31.29 \pm 0.75$ |
|  | BOIL + MOML | $41.22 \pm 0.83$ | $27.77 \pm 0.75$ | $32.98 \pm 0.79$ |
|  | CML (**ours**) | $39.52 \pm 0.80$ | $\mathbf{33.11 \pm 0.79}$ | $\mathbf{36.03 \pm 0.79}$ |
| 5-shot | MAML + CP | $56.16 \pm 0.72$ | $34.85 \pm 0.72$ | $42.91 \pm 0.71$ |
|  | MAML + MOML | $55.66 \pm 0.78$ | $39.38 \pm 0.77$ | $45.89 \pm 0.77$ |
|  | ProtoNet + CP | $59.11 \pm 0.71$ | $39.41 \pm 0.73$ | $46.93 \pm 0.71$ |
|  | ProtoNet + MOML | $58.72 \pm 0.74$ | $41.59 \pm 0.75$ | $48.59 \pm 0.74$ |
|  | BOIL + CP | $58.54 \pm 0.76$ | $34.28 \pm 0.75$ | $42.94 \pm 0.78$ |
|  | BOIL + MOML | $60.21 \pm 0.79$ | $35.47 \pm 0.78$ | $44.37 \pm 0.78$ |
|  | CML (**ours**) | $59.74 \pm 0.75$ | $\mathbf{49.48 \pm 0.76}$ | $\mathbf{54.01 \pm 0.74}$ |

accuracy, respectively. We compare the proposed constrained meta-learning (CML) with several benchmarks: (i) MAML [23] with constraint penalty (CP); (ii) ProtoNet [49] with CP; (iii) BOIL [44] with CP; (iv) MAML with MOML [57]; (v) ProtoNet with MOML; (vi) BOIL with MOML. As the baseline methods for few-shot learning, MAML, ProtoNet, and BOIL, can only deal with unconstrained meta-learning, at (i)(ii)(iii), we add a weighted penalty term of the robustness constraint violation to their loss functions. At (iv)(v)(vi), MAML, ProtoNet and BOIL fit into a multi-objective meta-learning (MOML) framework [57], which regards the clean accuracy and the PDG accuracy as two competing objectives. Since all baseline methods are offline meta-learning, for comparisons, we use the similar way shown in Algorithm 2 of Appendix B to adapt them to the online setting. Specifically, in each round, we sample a batch of tasks from the revealed tasks, and use the gradient-based method on their meta-objective functions over the data of sampled tasks. The modification from the offline baseline methods to their online versions are exactly following our approach in terms of optimization of the online meta-objective.

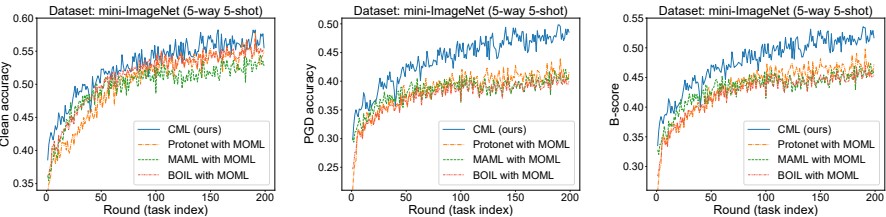

Figure 4: Test accuracy v.s. training task index on dataset mini-ImageNet. **Left**: Clean accuracy; **Middle**: PGD accuracy; **Right**: B-score.

We test the algorithms on two few-shot learning datasets, CUB [53] and mini-ImageNet [52]. Due to the page limit, the results on the CUB are shown in Appendix C.2. Table 2 shows that our method significantly improves the PGD accuracy and the B-score than the benchmarks and keeps the clean accuracy comparable. Fig. 4 shows that our method outperforms the benchmarks in terms of both test accuracy and learning speed, i.e., its test accuracy is larger than $0.5$ at $T = 30$, while the benchmarks take at least $50$ tasks to achieve so.

## 6   Conclusion

In this paper, we develop an online constrained meta-learning framework, which learns the meta-prior from learning tasks with constraints. We theoretically quantify the optimality gaps and constraint violations produced by the proposed framework. Both two metrics consider the generalization ability of the task-specific models to unseen data. Moreover, we propose a practical algorithm for the constrained meta-learning framework. Our experiments on meta-imitation learning and few-shot image classification demonstrate the superior effectiveness of the algorithm.

## Acknowledgments and Disclosure of Funding

This work is partially supported by the National Science Foundation through grants ECCS 1846706 and ECCS 2140175. We would like to thank the reviewers for their constructive and insightful suggestions.

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
