# OpenReview forum: "Online Constrained Meta-Learning: Provable Guarantees for Generalization"
_NeurIPS.cc/2023/Conference — NeurIPS 2023 spotlight_

### Official Review · Reviewer_MvUS · 2023-06-30

**Soundness:** 3 good
**Presentation:** 2 fair
**Contribution:** 3 good
**Rating:** 6
**Confidence:** 3

**Summary:**

The paper studies the problem of online meta-learning with constraints. After a formalization of the problem, the paper proposes an algorithm in the case where the loss function is convex, using Follow-the-Perturbed-Leader (FTPL) to update the meta-objective.
Then the paper theoretically prove upper bounds on the regret and the violation of the constraints for their constrained learning setting.
Finally, the paper presents experimental results on two different applications, meta imitation learning with collision avoidance and robust few-shot image classification.

**Strengths:**

### Originality

- The paper presents a detailed theoretical analysis in the constrained setting.
- The algorithm is a combination of iMAML [1] using FTPL instead of FTL for the online setting.

### Quality

- The paper presents and proves theoretical upper bounds in the constrained setting with their proposed algorithm, with a lot of theoretical developments in the appendix. It shows the soundness of their approach.

### Clarity

- The formalization of the problem is clear.

### Significance

- Results on the different benchmarks and applications presented show strong performance compared to baselines.

[1]: Rajeswaran, A., Finn, C., Kakade, S. M., & Levine, S. (2019). Meta-learning with implicit gradients. Advances in neural information processing systems, 32.

**Weaknesses:**

### Clarity

- The relation of the proposed algorithm to previous work is quickly dismissed. The authors introduce online meta-learning algorithms in l.23-33, and in the related works section (l.88 - 98), the paper presents only optimization-based meta-learning algorithm. However, there is no discussion to clearly explain the differences with their proposed algorithm.
- The way the tasks are set up in the applications are not very well described. Specifically in the robust few-shot classification benchmarks, it is not clear how the benchmark is adapted for this *online* setting, since these datasets are more commonly used for few-shot meta-learning.


### Quality

- If I understand correctly, the approach presented is a combination of iMAML with FTPL for the online setting and with the constraint penalty. Thus, it would make sense to add other meta-regularization algorithm baselines in the benchmarks, such as iMAML at least.
- In the robust few-shot classification application presented, the other baselines are not designed for *online* meta-learning. The comparison seems unfair.
- The authors state that their approach speeds up the adaptation to new tasks (l.357), but we can see in Figure 3 (right) that it is the opposite, their algorithm takes more time to adapt.


**Questions:**

I would like the authors to describe how the methods are evaluated on the robust few-shot image classification benchmarks:
- The models are trained on how many tasks ?
- Are the tasks different between meta-training and meta-testing ? If so, how many meta-testing tasks are considered ? Otherwise, I would like to see results of online meta-learning methods on this benchmark.
- It's not totally clear to me why the algorithm would be specific to the constrained case.

**Limitations:**

The paper does not include a discussion of the limitations of their approach. The algorithm presented seems to be more costly to run than online MAML.

---

> ### Author Rebuttal · Authors · 2023-08-04
>
> Thank you very much for your time and effort in reviewing our paper. We address your concerns as follows.
>
> >**Weakness of Clarity 1.**
>
> **Answer:** Sorry about the missing discussion. Here are the relation of the proposed algorithm to previous works and the intuition of our algorithm design. We will add them to our revised manuscript.
>
> In constrained meta-learning, the task-specific parameter should satisfy its given constraints.
> We employ the meta-regularization approach with hard constraints as the within-task algorithm, and use the follow-the-perturbed-leader to handle the meta-objective of sequential tasks.
> In the meta-initialization approaches shown in [38, 20, 21, 3, 32], as the within-task algorithm only takes a few optimization steps, even if we do optimization to reduce the constraint violation, the solution is far from feasible. On the other hand, the meta-regularization approaches shown in [15, 14, 43, 29] fully solve the within-task.
> To prioritize constraint satisfaction, we combine the meta-regularization approach with hard constraints for constrained meta-learning.
>
> >**Weakness of Clarity 2.**
>
> **Answer:** Sorry about the missing description. Here, we will show the experiment settings in the robust few-shot classification and how to modify them to the online setting.
>
> >> What are the online setting and the data setting for meta-training and meta-test.
>
> Take the mini-Imagenet dataset as the example (similar for the CUB dataset).
> The mini-Imagenet dataset includes 100 classes of images.
> We have 64 classes for training images, 16 classes for validation images, and 20 classes for test images.
> During the online meta-training, in each round, we sample a task only from the 64 training data classes and regard it as the revealed task, i.e., sample a 5-way k-shot task (5 classes and k images for each class). There are 200 rounds of online learning, and thus we sample 200 tasks from the training data. In the meta-test for Table 2, we use the test dataset with 20 classes. From the 20 test classes, we sample 600 tasks, i.e., 600 times of 5-way k-shot data from the 20 test classes, which means that the image classes in the 600 meta-test tasks are unseen in training tasks.
>
> >>How are the baseline methods adapted for this online setting.
>
> Since the baseline methods are offline, for comparisons, we use a similar way to Algorithm 2 (shown in Appendix A) to adapt them to the online setting. Specifically, in each round, we sample a batch of tasks from the revealed tasks, and use the gradient-based optimization method on their meta-objective functions over the data of sampled tasks. The modification from the offline baseline methods to their online versions exactly follows our approach in terms of optimization of the online meta-objective. Moreover, the dataset setting is the same for all baseline methods and our approaches.
>
> We will include the discussions in the revised manuscript.
>
> >**Weakness of Quality 3.**
>
> **Answer:** As the existing meta-regularization algorithms cannot handle neither the constraints for tasks nor online meta-learning (sequential task setting), we cannot directly compare our performances on the online constrained meta-learning problem.
>
> We compare our method with online-MAML. It is an online meta-learning method, and we add the constraint penalty loss to enable it to handle the constraints. As shown in the iMAML paper, the difference in the accuracy of MAML and iMAML is small (not larger than 1\%) in the two datasets. So, we think that online-MAML and an online version of iMAML will have comparable performance, and we only compare with online-MAML for convenience.
>
> >**Weakness of Quality 4.**
>
> **Answer:** As shown in response to weakness 2, since the baseline methods are offline, to do comparisons, we modify them to their online versions. The modification from the offline baseline methods to their online versions exactly follows our approach in terms of optimization of the online meta-objective. If it is the first time to study an offline learning problem, it is a standard practice to modify existing offline methods to solve the new online problem and compare the modified methods with newly developed methods [21][R2]. Thus, the comparisons are fair.
>
> [R2] Yao, Huaxiu, et al. "Online structured meta-learning."
>
> >**Weakness of Quality 5.**
>
> **Answer:** Sorry about the confusion. We want to claim that "Figure 3 shows that our method achieves a comparable adaptation time to with online-MAML while outperforming online-MAML in terms of test error and collision avoidance." We will modify the statement in our revised manuscript.
>
> >**Question 1. The models are trained on how many tasks.**
>
> **Answer:** As shown in response to weakness 2, the mini-Imagenet dataset includes 100 classes of images. We have 64 classes for training images, 16 classes for validation images, and 20 classes for test images. From the 64 training image classes, we sample 200 tasks (5-way k-shot learning tasks) for meta-training. From the 20 test image classes, we sample 600 tasks for the meta-test. So the test tasks used to test the performance are unseen in the meta-training phase.
>
> >**Question 2. Are the tasks different between meta-training and meta-test? How many meta-test tasks are considered?**
>
> **Answer:** As shown in responses to Weakness 2 and Question 1, the tasks between meta-training and meta-test are different. The test tasks used to test the accuracy (Table 2) are unseen in the meta-training phase. We have 600 tasks sampled from the test dataset as the meta-test tasks.
>
> >**Question 3. Why the algorithm would be specific to the constrained case.**
>
> **Answer:** Algorithm 2 (in Appendix A) uses the primal-dual approach to solve constrained optimization in Equation (3) (line 3 of Algorithm 1), and uses the constrained bilevel optimization analysis [50] to compute the gradient of the constrained bilevel objective function in Equation (4) and minimize the function (line 6 of Algorithm 1).

---

> > ### Comment · Reviewer_MvUS · 2023-08-14
> >
> > Thanks for the detailed answer and for the clarifications. I'm satisfied with the authors' answer, I don't have any more concerns and raise my score. I encourage the authors to add these discussions in the revised version.

---

### Official Review · Reviewer_jLc8 · 2023-07-03

**Soundness:** 3 good
**Presentation:** 3 good
**Contribution:** 3 good
**Rating:** 8
**Confidence:** 4

**Summary:**

In this paper, a novel online constrained meta-learning framework is presented. The framework is designed to facilitate continuous learning from sequential tasks while ensuring that these tasks adhere to strict constraints. In addition to existing analyses of meta-learning, this study goes further by presenting the upper bounds for optimality gaps and constraint violations that arise from the proposed framework.
This framework takes into account the dynamic regret of online learning and the generalization capability of the task-specific models. Finally, the paper offers a practical algorithm to implement the framework, and its superior effectiveness is validated through experiments conducted in the domains of meta-imitation learning and few-shot image classification.

**Strengths:**

- This paper consider that the meta-objective is non-convex.
- Study the dynamic regrets.
- Two elaborate applications demostrate the superior effectiveness of the algorithm.

**Weaknesses:**

- The bound is scaled with $\mathcal{O}(\frac{1}{\sqrt{T}})$. This seems to ignore the size of every training datasets.
- The comparion with existing online meta learning bounds is necessary.

**Questions:**

- In section 2.1, the proposed constrained optimization paradigm in the paper requires that different tasks satisfy certain constraints on errors under any task-adaptive parameters. Intuitively, this may seem contradictory to improving the performance of specific tasks. Task-adaptive parameters should ideally be focused solely on specific tasks, so why is there a need to ensure the performance of other tasks simultaneously? Is it more reasonable to constrain the meta-parameters?

**Limitations:**

 It should compare with existing online meta learning bounds.

---

> ### Author Rebuttal · Authors · 2023-08-04
>
> Thank you very much for your time and effort in reviewing our work. Thanks for your suggestions. We address your concerns as follows.
>
> > **Weakness 1. The bound is scaled with $\mathcal{O}\left(\frac{1}{\sqrt{T}}\right)$. It seems to ignore the size of training datasets.**
>
> **Answer:** The upper bound in Theorem 1 is
> $\mathcal{O}\left(\mathcal{S}^*\left(\mathcal{T}\_{1: T}\right) \sqrt{\frac{\ln \left|\mathcal{D}\_0^{t r}\right|}{\left|\mathcal{D}\_0^{t r}\right|}}+\sqrt{\frac{\ln \left|\mathcal{D}\_{+}^{t r}\right|}{\left|\mathcal{D}\_{+}^{t r}\right|}}+\sqrt{\frac{\ln \left|\mathcal{D}\_{0}^{val}\right|}{\left|\mathcal{D}\_{0}^{val}\right|}}+\frac{1}{\sqrt{T}}\right)$.
> As shown in Appendices E and F, the coefficients of the notations $\mathcal{O}$ are independent of the size of training datasets and only depend on some constants, such as the Lipschitz constant $L_0$.
>
> * The first term $\left(\mathcal{S}^*\left(\mathcal{T}\_{1: T}\right) \sqrt{\frac{\ln \left|\mathcal{D}\_0^{t r}\right|}{\left|\mathcal{D}\_0^{t r}\right|}}+\sqrt{\frac{\ln \left|\mathcal{D}\_{+}^{t r}\right|}{\left|\mathcal{D}\_{+}^{t r}\right|}}+\sqrt{\frac{\ln \left|\mathcal{D}\_{0}^{val}\right|}{\left|\mathcal{D}\_{0}^{val}\right|}}\right)$ depends on the size of training datasets. As shown in Equation (5) and Proposition 2, the term quantifies the generalization error when limited training data is given.
>
> * The last term $\left(\frac{1}{\sqrt{T}}\right)$ is independent of the size of training datasets.
> The term $\mathcal{O}\left(\frac{1}{\sqrt{T}}\right)$ is the gap of the meta-objective function $
> \sum_{t^{\prime}=1}^{t} \mathcal{L}^{val}(\mathcal{A}lg(\lambda,\phi, \mathcal{D}\_{t^{\prime}}^{tr}),\mathcal{D}\_{0,t^{\prime}}^{val} )
> $ (defined in line 193) between the meta-parameter $\phi=\phi_t$ produced by our algorithm and the optimal meta-parameter $\phi=\phi^*$.
> The size of the training dataset will not influence the gap of the meta-objective function values between $\phi_t$ and $\phi^{\*}$, because the values on both $\phi_t$ and $\phi^*$ use $\mathcal{A}lg$ in Equation (3) with the same training data (limited data size) to obtain the solutions. No matter whether the size of the training data is large or small, the size is shared for $\phi\_t$ and $\phi^{\*}$ and imposes the same error on $\phi\_t$ and $\phi^{\*}$.
>
> > **Weakness 2. Comparison with existing online meta-learning bounds.**
>
> **Answer:** In Table 1, we compare the used metrics in our paper and existing works [3] [14] [1] [21]. The metrics used in this manuscript are harder to quantify than most existing papers, in terms of constraint, generalization, and dynamic regret. Here, we compare our results with [3] [14] [1] [21] [R1].
>
> * We consider the constraints in each learning task, and thus need to (a) quantify the constraint violations and (b) quantify the error on the loss function introduced by the inexact constraint approximation. These are not considered in all existing works.
> * If all the constraints are removed from our problem, our result $\mathcal{O}(\mathcal{S}^{\*}(p(\mathcal{T})) \sqrt{\frac{\ln{|\mathcal{D}\_{0}^{tr}|}}{|\mathcal{D}\_{0}^{tr}|}}  +\frac{1}{\sqrt{T}})$ has the same order as the upper bound ${\mathcal{O}}(\ln(n) / \sqrt{n})+\mathcal{O}(1 / \sqrt{T})$ shown in [R1] and [14], in terms of the number of tasks $T$ and the number of the within-task data points $n$ or $|\mathcal{D}_{0}^{tr}|$.
> * Paper [3] does not consider the generalization error produced by the limited data size, and considers a strongly-convex meta-objective (ours is non-convex). Then, the bound is $\mathcal{O}(\mathcal{S}^{*}(p(\mathcal{T}))+\frac{1}{{T}})$, which is independent of $|\mathcal{D}_{0}^{tr}|$ and has the order of $\mathcal{O}(\frac{1}{{T}})$ because of the strongly convexity.
> * Paper [1] consider a strongly-convex meta-objective (ours is non-convex) and a static regret (ours is a dynamic regret) and has the order of $\mathcal{O}(\frac{1}{{T}})$.
> * In conclusion, for the degenerate case where the constraints are removed, our bound has the same order as the state-of-art works. If we further impose stronger assumptions (such as strong convexity) or consider a simpler metric (such as generalization ignored), the bound could be better.
>
> We will include the above discussion in the revised manuscript. Here is the reference.
>
> [R1] Denevi et al., 'Online-Within-Online Meta-Learning'.
>
> > **Question 1. In section 2.1, the proposed constrained optimization paradigm in the paper requires that different tasks satisfy certain constraints on errors under any task-adaptive parameters. Intuitively, this may seem contradictory to improving the performance of specific tasks. Task-adaptive parameters should ideally be focused solely on specific tasks, so why is there a need to ensure the performance of other tasks simultaneously?**
>
> **Answer:** In section 2.1 and other sections of the manuscript, the task-specific parameter always only needs to satisfy the constraints for its specific task, and doesn't require that different tasks satisfy certain constraints.
>
> In section 2.1, the task $\mathcal{T}_t$ is characterized by its data distributions $\mathcal{D}\_{t}=\{\mathcal{D}\_{0,t},\mathcal{D}\_{1,t}, \ldots, \mathcal{D}\_{m,t}\}$. Here, $\mathcal{D}\_{i,t}$ is the constraint dataset only for task $\mathcal{T}_t$, the constraint satisfaction of $\mathbb{E}\_{z \sim \mathcal{D}\_{i,t}}\left[\ell_i(\theta,z)\right] \leq c\_{i,t}$ with $\mathcal{D}\_{i,t}$ is specific for task $\mathcal{T}\_t$. From Equation (1), different tasks (e.g., $\mathcal{T}\_{t_1}$ and $\mathcal{T}\_{t_2}$) have different constraint datasets ($\mathcal{D}\_{i,t_1}$ and $\mathcal{D}\_{i,t_2}$ for $i$-th constraint), and should satisfy different constraint functions ($\mathbb{E}\_{z \sim \mathcal{D}\_{i,t_1}}\left[\ell_i(\theta,z)\right] \leq c\_{i,t_1}$ and $\mathbb{E}\_{z \sim \mathcal{D}\_{i,t_2}}\left[\ell_i(\theta,z)\right] \leq c\_{i,t_2}$).

---

> > ### Comment · Reviewer_jLc8 · 2023-08-12
> > **Thank you for your response**
> >
> > Thanks for your detailed response. Could you provide more details on the question "Is it more reasonable to constrain the meta-parameters?"

---

> > > ### Author Response · Authors · 2023-08-12
> > > **Answer to the question ""Is it more reasonable to constrain the meta-parameters?"**
> > >
> > > Each task $\mathcal{T}\_t$ is characterized by its loss function on the data distribution $\mathcal{D}\_{0,t}$ and its constraint functions on the constraint data distributions $\{\mathcal{D}\_{1,t}, \ldots, \mathcal{D}\_{m,t}\}$. The task-specific parameter $\theta^{\prime}\_{t}$ for task $\mathcal{T}\_t$ needs to satisfy its task-specific constraints. There is no common constraint shared by all tasks, and it is not reasonable to impose constraints on the meta-parameter.

---

> > > > ### Comment · Reviewer_jLc8 · 2023-08-16
> > > > **Thanks for your further response**
> > > >
> > > > Thanks for the detailed explanations. I tend to increase my score to 8.

---

### Official Review · Reviewer_CTub · 2023-07-05

**Soundness:** 4 excellent
**Presentation:** 2 fair
**Contribution:** 3 good
**Rating:** 7
**Confidence:** 4

**Summary:**

The paper studies the theory of biased-regularization meta-learning under the sequential task setting. Despite there having been previous works in this area, this paper distinguishes itself by introducing the concept of the Online Constrained Meta-Learning problem and presenting a straightforward solution. It applies constrained optimization with biased meta-regularization, utilizing Follow-the-Perturbed-Leader (FTPL) FTPL to handle the non-convex meta-objective function, providing theoretical analysis and proofs of upper bounds, and developing a practical algorithm for large-scale problems. Empirical experiments validate the effectiveness of the proposed algorithm in meta-imitation and meta-reinforcement learning.

**Strengths:**

- This paper first studies the online constrained meta-learning, which is rarely concerned. To this end, this paper gave the formal problem formulation of constrained sequential learning.

- The author distinguishes the assumptions proposed by this paper and follows others. It makes it easier to examine the assumptions applied in this paper.

- A solid work that follows the setting of learning with biased regularization. The author gives a detailed discussion of different cases.

**Weaknesses:**

- Of a particular relevant missing work, [1] can also be deemed as the constrained (conditional) meta-learning. The author should further discuss this work since [1] also shows the generalization results.

- More implications are needed to explain the results, i.e., Corollary 1.

- The bounds of the derived results have not been examined. For instance, the second term in RHS of Prop 3 is the order of $\mathcal{O}(d^2\mathcal{B} L_0^2 \ln |\mathcal{D}^{tr}_0|)$, which can be a dominate term.  Furthermore, the diameter $\mathcal{B}$ and model size $d$ can also be large enough. This result may not be applied to overparameterized settings.

- The relationship between $\mathcal{D}_t$ and $\mathcal{D}$ is vague. Suppose $\mathcal{D}$ refers to any $t$ in $\mathcal{D}_t$. The statement of Propositions and Theorems should clarify this point.

- From the Proofreading of Appendix 4, I found the bounds in Proposition 3 & 4 depend on many terms. However, in Prop.1 & 2, the author removes the small terms (in Big-O notation) without discussing the orders of these terms in what limits.

——Minors——
- Confusing statement about “Problem (1), (3)”. Since the author refers to Equations (1) and (3) as “Algorithms” and “Problems” simultaneously, it may be better to define the “Problems” officially.

- Adding a notation cheatsheet in the appendix may be more readable.

**Questions:**

- Why define $D(\phi, \mathcal{T}_{1:T})$, $\mathcal{S}^*(\rho(\mathcal{T}))$ in square root form?

- What will happen if we only learn the same task, i.e., $\mathcal{S}^*(\rho(\mathcal{T})) \to 0$, then the coefficient will be infinitely large $\lambda \to \infty$. Do the results still hold true in such a degenerate case?

- Is Assumption 1 should hold $\forall t$? If it is, please make it clear.

- As mentioned above, what does big-O notation mean in the results?

**Limitations:**

N/A pure theoretical work.


**References**:

[1] Denevi, Giulia, Massimiliano Pontil, and Carlo Ciliberto. "The advantage of conditional meta-learning for biased regularization and fine-tuning." Advances in Neural Information Processing Systems 33 (2020): 964-974.

---

> ### Author Rebuttal · Authors · 2023-08-02
>
> Thank you very much for your time and effort in reviewing our work. Thanks for your suggestions and reference recommendation. We address your concerns as follows.
>
> > **Weakness 1.  Discussion of connections with conditional meta-learning.**
>
> **Answer:** Thanks for the reference. Constrained meta-learning and conditional meta-learning aim to achieve different goals.
>
> In meta-learning, the goal of the within-task is minimizing the expected loss.  Meta-learning aims to learn a shared meta-parameter for all tasks that can improve the learning of new tasks.
>
> In conditional meta-learning, the goal of the within-task is minimizing the expected loss, which is the same as meta-learning. Different from meta-learning, conditional meta-learning aims to learn a map from the task information to its meta-parameter to facilitate within-task learning, rather than learning a shared meta-parameter for all tasks in meta-learning.
>
> In our constrained meta-learning, the goal of the within-task is minimizing the expected loss while satisfying imposed constraints. The constraints are not included in meta-learning and conditional meta-learning. The approaches of solving them are also different.
>
> > **Weakness 2.  More implications of Corollary 1.**
>
> **Answer:** Here is the implication of Corollary 1. We will include the discussion in the revised manuscript.
>
> Corollary 1 considers that the revealed tasks are sampled from a static task distribution. In this case, when $T$ is sufficiently large, the online meta-learning algorithm degenerates to an algorithm for offline meta-learning. If we further ignore the constraints in our problem setting, the bound of Corollary 1 holds the same order as the upper bound shown in [14].
>
> > **Weakness 4. The relationship between $\mathcal{D}\_t$ and $\mathcal{D}$.**
>
> **Answer:** Sorry about the confusion. We denote ${D}(\phi, \mathcal{T}\_{1:T})$ as the parameter distance, denote $\mathcal{D}\_t$ as the data distribution for task $\mathcal{T}\_t$, and denote the edge length as $D$. These notations are too close. We will modify ${D}(\phi, \mathcal{T}\_{1:T})$ to $\mathcal{D}ist(\phi, \mathcal{T}\_{1:T})$, and modify $D$ to $D\_l$ in the revised manuscript.
>
> > **Weakness 5 and Question 4. The Big-O notation with respect to what limits/ what does big-O notation mean in the results.**
>
> **Answer:** We consider the Big-O notation only with respect to the limits of (i) the data numbers including $\|\mathcal{D}\_{0}^{tr\}|$, $\|\mathcal{D}\_{0}^{val\}|$ and $|\mathcal{D}\_{+}^{tr}|$ (ii) the task similarity $\mathcal{S}^{\*}( \mathcal{T}\_{1:T})$ (iii) the task number $T$.
>
> > **Weakness 6.  Refer about Equations (1) and (3) as “Algorithms” and “Problems” simultaneously.**
>
> **Answer:** Sorry about the confusion. Equation (1) is a problem. Equation (3) is an algorithm to approximate the solution of Equation (1). Equation (3) includes an optimization problem. We will clarify them in the revised manuscript.
>
> > **Weakness 7. Adding a notation checksheet in the appendix may be more readable.**
>
> **Answer:** Thank you for the suggestion. We attach the notation list in the global rebuttal PDF file, and will add it to the revised appendix.
>
> > **Question 1.  Why define $D\left(\phi, \mathcal{T}\_{1: T}\right), \mathcal{S}^{\*}(\rho(\mathcal{T}))$ in square root form.**
>
> **Answer:** Following [14], we would like to define $D\left(\phi, \mathcal{T}\_{1: T}\right)$ and $\mathcal{S}^*(\rho(\mathcal{T}))$ as metrics of the parameter distance, and thus consider the square root of the quadratic sum $\frac{1}{T}\sum_{t=1}^T \frac{1}{2}\|\|{\theta}^{\*}_t-{\phi}\|\|^2$.
>
> > **Question 2.  What will happen if we only learn the same task, i.e., $\mathcal{S}^{\*}(\rho(\mathcal{T})) \rightarrow 0$, then the coefficient will be infinitely large $\lambda \rightarrow \infty$. Do the results still hold true in such a degenerate case?**
>
> **Answer:** The result still holds in the degenerate case. Here is the reason.
>
> When $\lambda \rightarrow \infty$, the term $\frac{\lambda}{2}\|\|\theta-\phi_t\|\|^2$ totally dominates in the objective function of the optimization problem of Equation (3), then the problem reduces to
>
> ${\theta}\_t = \mathcal{A}lg(\lambda,\phi\_t, \mathcal{D}\_{t}^{tr})=\arg\min_\theta \|\|\theta-\phi_t\|\|^2, s.t. \frac{1}{|\mathcal{D}\_{i,t}^{tr}|} \sum\_{z \in \mathcal{D}\_{i,t}^{tr}}\ell_i(\theta,z) \leq c\_{i,t}, \ i=1, \ldots, m. $
>
> As we can see, the the solution ${\theta}\_t$ will not depend on $|\mathcal{D}\_{0}^{tr}|$ and only depend on $|\mathcal{D}\_{+}^{tr}|$, $|\mathcal{D}\_{0}^{val}|$, and $T$. This result corresponds to Theorem 1 with $\mathcal{S}^*(\rho(\mathcal{T})) \rightarrow 0$, where the result will not depend on $|\mathcal{D}\_{0}^{tr}|$ and only depend on $|\mathcal{D}\_{+}^{tr}|$, $|\mathcal{D}\_{0}^{val}|$, and $T$. So the results still hold in such a degenerate case.
>
> > **Question 3.  Is Assumption 1 should hold for all $t$ ?**
>
> **Answer:** Yes, we will clarify it in the revised manuscript.

---

> > ### Comment · Reviewer_CTub · 2023-08-14
> >
> > Thanks for the detailed explanations of my questions. I have decided to increase the score since the authors address my concerns.

---

### Official Review · Reviewer_FYCS · 2023-07-06

**Soundness:** 3 good
**Presentation:** 2 fair
**Contribution:** 2 fair
**Rating:** 6
**Confidence:** 3

**Summary:**

The authors propose an online constrained meta-learning algorithm that is able to sequentially learn a  sequence of tasks that are subject to hard (and stochastic) constraints. The authors also theoretically quantify the optimality gaps and constraint violations produced by the proposed method, by considering the dynamic regret of online learning and the generalization ability of the task-specific models. They also validate the effectiveness of the proposed method on numerical experiments on meta-imitation learning and few-shot image classification.

**Strengths:**

The authors validate their proposed method, both theoretically and experimentally.

The authors present a new meta-learning method aiming at facing the more challenging situation in which the tasks are stochastically subject to constraints.

**Weaknesses:**

The statements and the notation of the paper could be simplified and made more intuitive, less heavy.

Why can the constrained meta-learning framework be interesting in practical application? The authors do not well motivate the setting they consider. This is a very important aspect in my opinion. Could you please describe some examples of possible applications in which the proposed constrained setting can be useful/necessary?



**Questions:**

The constraints are random variables. This seems to be problematic. Which is the main trick you use to deal with it?

The authors use the notion of variance of the task optimal parameters to measure the similarity among the tasks, as in [14] and the paper [A] I mention here below. Could you please make a clearer comparison between your rates and those obtained in these other work, by just looking at the main important constants and leading terms w.r.t. the number of tasks and within-task points? Consider also the different assumptions.

[A] Denevi et al., 'Online-Within-Online Meta-Learning'.



**Limitations:**

I do not see any potential negative societal impact related to this work.

---

> ### Author Rebuttal · Authors · 2023-08-04
>
> Thank you very much for your time and effort in reviewing our work. Thanks for your suggestions and reference recommendation. We address your concerns as follows.
>
> > **Weakness 1.  The statements and the notation of the paper could be simplified and made more intuitive.**
>
> **Answer:** Thank you for the suggestion. We will simplify the notations and attach the notation list in the global rebuttal PDF file, and will add it to the revised appendix.
>
> > **Weakness 2.  Why can the constrained meta-learning framework be interesting in practical application and what are the motivation examples.**
>
> **Answer:** Constrained meta-learning learns a meta-parameter from existing constrained learning tasks, where each task needs to minimize its expected loss while satisfying its constraints.
> When a new constrained learning task is revealed, the meta-parameter is adapted to the new task, which improves the learning efficiency and reduces the expected risk and constraint violation on the new tasks.
> Existing meta-learning approaches can only learn from unconstrained learning tasks.
>
> Below are a couple of examples that motivate the constrained meta-learning framework.
>
> * One motivated example is imitation learning with collision avoidance in changing environments, as shown in our first experiment (Section 5.1). The expert performs demonstrations in a free space. The learner can observe the demonstrations and is asked to perform the task in a new cluttered environment quickly. The new environment is uncertain and unknown to the learner until the task is revealed.
>
> * Another motivated example is robot control with different dynamics. Consider a scenario that a robot needs to quickly deploy a new control policy once its dynamics change. The problem can be formulated as constrained meta-learning, where actuation limitations are imposed as constraints for the control policy. The within-task is to find the task-specific control policy that can minimize the control cost for the robot while satisfying the actuation constraints of the robot dynamics. The meta-algorithm is to optimize a meta control policy that adapts to the task-specific control policy once the new dynamics is given.
>
> We will include the above discussion in the revised manuscript.
>
> > **Question 1.  The constraints are random variables. This seems to be problematic. Which is the main trick you use to deal with it?**
>
> **Answer:**
>
> * In constrained stochastic optimization, it is standard to define both the objective function and the constraint functions as functions of random variables [7][16].
> * In our proposed algorithm, we use the sample average approximation method to approximate the expectation, i.e., using the empirical average of the constraint functions on the given training data to approximate the constraint function defined by the whole data distribution, as shown in Equation (3) (line 176 in Section 3).
> * As shown in Proposition 4 of Appendix D and Appendix C.2, we quantify the error between the sample average approximation and the expectation by analyzing the Rademacher complexity [7][16] of the constraint functions.
>
> > **Question 2.  Comparison between your rates and those obtained in the related works [A][14].**
>
> **Answer:**
>
> **Comparison of assumptions.** Papers [A] and [14] assume the learning model is linear, i.e., $\mathcal{E}(Z)=\sum_{i=1}^n \ell_i(\langle x_i, w_i\rangle)$, while ours could be any function, such as a neural network. Our paper, [14] and [A] have similar assumptions about the convexity of functions. By adding a regularizer in the meta-objective function, the overall meta-objective in [14][A] is convex, while the meta-objective function in our paper is non-convex.
>
> **Comparison of metrics.** Paper [14] considers offline meta-learning and quantifies the expected optimality gap of the task-specific parameters over the task distribution. Paper [A] considers online meta-learning and the dynamic regret of the expected optimality gaps over sequential tasks. Similar to [A], we consider the dynamic regret of the expected optimality gaps.
> Beyond [A], we consider the constraints, and thus need to (a) quantify the constraint violations and (b) quantify the error on the loss function introduced by the inexact constraint approximation.
>
> **Comparison of rates.** If all the constraints are removed from our problem, our result has the same order as the upper bound shown in Corollary 7 of [A] and Theorem 6 of [14], in terms of the number of tasks $T$ and within-task points $n$.
>
> We will include the above discussion in the revised manuscript.

---

> > ### Comment · Reviewer_FYCS · 2023-08-17
> > **Response to the authors**
> >
> > I thank the authors for their response. They answered my questions.

---

### Author Rebuttal · Authors · 2023-08-04

We are grateful and indebted for the time and effort invested to evaluate our manuscript by all reviewers, and for all the suggestions and reference recommendations to make our manuscript a better and stronger contribution. Please find below our detailed replies to all the comments of the reviewers.

We notice that some reviewers suggest to attach a notation checklist. We attach a notation list in the global rebuttal PDF file for the convenience of our discussion and also add it to the revised appendix.

---

### Decision · Program_Chairs · 2023-09-21

**Decision:**

Accept (spotlight)

**Comment:**

This paper provides original contribution in online constrained meta-learning under the sequential task setting. The paper proposes further analysis with respect to state of the art by deriving theoretical results in the form of upper bounds for optimality gaps and constraint violations. The proposed framework takes into account the dynamic regret of online learning and the generalization capability of the task-specific models. Some effective experimental evaluation meta-imitation learning and few-shot image classification are presented.

Reviewers recognized the quality and interestingness of the paper.
Some issues about the notations, some statements, and precisions with respect to other methods and settings were raised by the reviewers.
During the rebuttal, authors have provided explanations to the points of the reviewers, they also joined a notation checklist about the notations used.
All reviewers have acknowledged that their concerns have been addressed.

Based on all the proviso remarks I propose then acceptance.
I recommend to the authors to insert the notation checklist in the revised appendix of the paper, in line with their commitment, and to revise the paper by including all the precisions and discussions evoked during the discussion with reviewers.